# Gesture recognition with Brownian reservoir computing using geometrically confined skyrmion dynamics

Grischa Beneke [1], Thomas Brian Winkler [1], Klaus Raab [1],
Maarten A. Brems [1], Fabian Kammerbauer [1], Pascal Gerhards [2],
Klaus Knobloch[2], Sachin Krishnia[1], Johan H. Mentink[3] & Mathias Kläui[1,4] ✉

Physical reservoir computing leverages the dynamical properties of complex physical systems to process information efficiently, significantly reducing training efforts and energy consumption. Magnetic skyrmions, topological spin textures, are promising candidates for reservoir computing systems due to their enhanced stability, non-linear interactions and low-power manipulation. Traditional spin-based reservoir computing has been limited to quasi-static detection or real-world data must be rescaled to the intrinsic timescale of the reservoir. We address this challenge by time-multiplexed skyrmion reservoir computing, that allows for aligning the reservoir's intrinsic timescales to real-world temporal patterns. Using millisecond-scale hand gestures recorded with Range-Doppler radar, we feed voltage excitations directly into our device and detect the skyrmion trajectory evolution. This method scales down to the nanometer level and demonstrates competitive or superior performance compared to energy-intensive software-based neural networks. Our hardware approach's key advantage is its ability to integrate sensor data in real-time without temporal rescaling, enabling numerous applications.

Skyrmions are chiral magnetic whirls that have been shown to exhibit enhanced stability due to their non-trivial topology[1–4]. They have shown great potential in non-conventional computing devices[5–13], or as information carriers in novel data storage[3,14–16]. Stabilized due to the Dzyaloshinskii-Moriya interaction, they can be present in bulk systems[17], or in thin film systems[2,18–20], where they exhibit particle-like behavior[3,5]. Proposals for applications exploit the unique characteristics of skyrmions for instance as bit-like information carriers[14,15], and use deterministic motion, controlled nucleation and annihilation of the spin textures for deterministic memory operations[21]. On the other hand, stochastic (thermal) dynamics is exploited for Brownian computing approaches[5,8,13,22,23]. In Brownian computing, the computation speed is tied to the systems' diffusion coefficient and can thus be tuned[8,24]. Therefore, skyrmion systems are particularly favorable as

skyrmions can exhibit significant thermal diffusion at room temperature[5] and diffusion-tuning mechanisms have been developed recently that allow for tuning the thermal dynamics over many orders of magnitude[24]. In both deterministic and stochastic applications, the manipulation of skyrmions can be achieved efficiently even with ultra-low current densities through currents that generate spin transfer torques[25,26] and spin-orbit torques (SOT)[27,28], as well as external fields[29], strain[30] and temperature gradients[31]. In thermally-activated skyrmion systems, pinning effects can be overcome by thermal excitations, allowing for directed motion at even lower current densities[5,22,32], where ultra-low currents bias the diffusive dynamics[22]. Thereby, the Brownian computing paradigm can offer low-power computing while simultaneously overcoming reproducibility issues due to variations between devices.

[1]Institut für Physik, Johannes Gutenberg-Universität Mainz, Mainz 55099, Germany. [2]Infineon Technologies Dresden, Dresden 01099, Germany. [3]Radboud University, Institute for Molecules and Materials, Nijmegen 6525, the Netherlands. [4]Center for Quantum Spintronics, Norwegian University of Science and Technology, Trondheim 7491, Norway. ✉e-mail: Klaeui@uni-mainz.de

Physical Reservoir Computing (RC) represents a key machine learning approach, where a non-linear physical reservoir is harnessed to process and map input data into a higher-dimensional state-space where they become for instance linearly seperable[33]. By performing a measurement of the reservoir, the complex state is subsequently mapped to an output state of measurement values. Using this concept, one can potentially reduce a highly complex problem to a linear one, given the dynamics of the system are suited to process the signal properly. This has a major advantage compared to deep recurrent neuronal networks where all the weights are trained, which is complex, slow and energy inefficient. In RC only the output weights are trained while the reservoir is fixed, resulting in faster and lower-energy operation[34]. Further, as the computation is performed by the systems' efficient intrinsic dynamics, the power consumption of such a device is typically much lower than inferring a software-based solution with similar performance.

Reservoir computing has been studied in magnetic systems[9–13,22,35–37] for static spatially multiplexed pattern recognition[22] and for the recognition of dynamic time-varying signals using spin structure dynamics[9], spin torque oscillators[38] and spin waves[35], which have intrinsic timescales in the MHz to GHz regime. While a proof-of-concept has been demonstrated for vowel recognition[39], the involved intrinsic magnetic dynamic frequencies in the MHz–GHz range require a complex and energy-intensive rescaling of the timescales of the spoken language or many other real-world signals that are to be recognized.

In this context, the most conspicuous applications for dynamic pattern recognition, ranging from speech to the detection of motion, involve such dynamics occurring on the microsecond (µs) to second (s) timescales. However, this is orders of magnitude slower than the spin dynamics, conventionally used for reservoir computing, which requires cumbersome and energy-intensive rescaling of the timescales.

Here, we employ thermally activated diffusion and current-induced displacement of skyrmions in a reservoir to detect real human gestures. We input the Doppler radar data of gestures as ultra-low power current-driven dynamics that in combination with geometrical confinement results in an intrinsic zero-power reset mechanism. Having comparable time scales of the Doppler radar data and the intrinsic dynamics and thus processing speed of our reservoir enables direct feeding of the sensor data into the reservoir. We find that the radar data of different hand gestures is detected in our hardware reservoir with a fidelity that is at least as good as a state-of-the-art software-based neural network approach.

## Results

Our device consists of a $Ta(5)/Co_{20}Fe_{60}B_{20}(0.95)/Ta(0.09)/MgO(2)/Ta(5)$ multilayer stack (thicknesses of layers are given in nanometers in parentheses) that is structured into an equilateral triangular geometry, with a side length of 36 µm. Figure 1 schematically depicts such a device. Such geometrical confinement was shown to lead to an equilibrium position of the skyrmion in the center[16,40–42] combined with non-linear current-driven skyrmion displacements due to spin-orbit torques[22,27,28,43–46]. The thermally activated diffusive dynamics results in stochastic motion, due to the low pinning of the multilayer stack[5,32,47] allowing for displacement at very small current densities and leading to a zero-power reset mechanism when the current is switched off. To tune the speed of the dynamics, an additional oscillating out-of-plane magnetic field component is used, that was shown to tune the diffusion speed over orders of magnitude[24].

The skyrmion is nucleated by using an in-plane magnetic field pulse, while an additional constant out-of-plane magnetic field stabilizes the skyrmion[5,22,24]. After the skyrmion is nucleated, the edge repulsion along with thermal dynamics causes the skyrmion to reside at the center of the device. Consequently, whether upon initialization

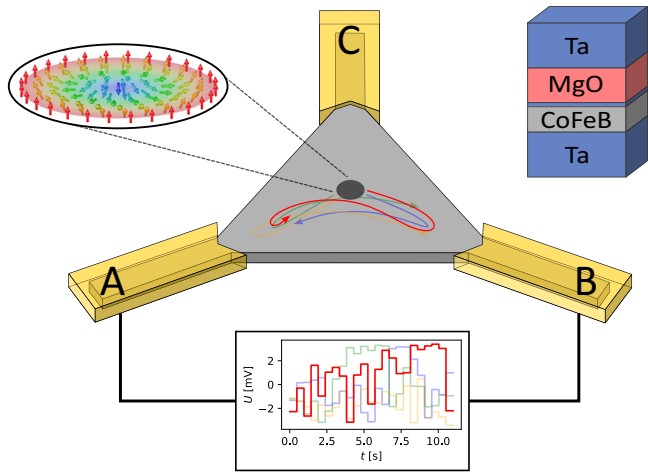

**Fig. 1 | Schematic representation of the dynamic Brownian reservoir computer.** The triangular stack (light gray; composition shown in the top right) is connected with chromium/gold contacts at the corners (half transparent yellow). In the used setup, only the lower corners A and B are connected to the time-dependent voltage. Here, four examples are shown in different colors, from each gesture type one input signal. The third corner C is not connected and therefore floating. The skyrmion is presented as a dark gray spot as visualized by magneto-optical Kerr microscopy. Possible skyrmion response trajectories to the time dependent voltages are shown by colored arrows.

or subsequent excitation through current-induced motion, the device inherently reverts to its original ground state due to this automatic reset mechanism[22,42].

To operate as a time-multiplexed reservoir, different signals in the form of time-dependent voltages are applied to the device. This results in a time-dependent displacement of the skyrmion. As the magnitude of the applied voltage increases, and subsequently the current density and thus SOT, the skyrmion position is pushed closer towards the corners of the device. This relationship is nonlinear due to the non-linearity of the boundary interaction potential in our system[48], as well as due to non-uniform current density imposed by the device geometry. The skyrmion is driven by very low current densities up to $2.9 \times 10^7 \, Am^{-2}$ at half width of the triangle. This current density is lower than any previously demonstrated displacement in multilayer skyrmion stacks as realized by the artificial mitigation of pinning effects by the oscillating out-of-plane magnetic field[24]. Stronger current densities may result in the annihilation of the skyrmion, while low current densities cause motion indistinguishable compared to purely thermal movement. The amplitude of the input signal is scaled within the range of negative and positive maximum voltage, enabling displacement of the skyrmion to either the left or right corner of the device (see methods section for more details). After finishing the input of the radar signal data, the skyrmion relaxes back to the center.

With this procedure, every possible time dependent voltage can be mapped to a set of time dependent movements of the skyrmion. Due to the super-imposed motion caused by current induced SOT and thermally activated diffusion, which enables skyrmions to overcome pinning sites, the skyrmion response will not be identical, but similar enough for the reservoir to function reliably. An essential requirement of reservoir computing is the nonlinear reservoir. The presented skyrmion reservoir is non-linear in multiple aspects, the skyrmion velocity being dependent on the current[45], the current density being dependent on the location, the skyrmion edge repulsion[48], and the skyrmion diffusion.

In our proof-of-concept experiment, the time-dependent position of the skyrmion is captured using a magneto-optical Kerr-effect (MOKE) microscope, which records the full information about the

position of the skyrmion at a framerate of 16 Hz. In real applications, a single time-dependent readout of a magnetic tunnel junction (MTJ) would be used, which is emulated here by reading the data from a fixed $0.5\,\mu m$ diameter area that is positioned in the device. A detailed explanation of the conversion process from Kerr images to MTJ signals is provided in the methods section and is depicted in Fig. 2a, b. Moreover, depending on the input timescale, we reduce the time resolution by using only every $n^{th}$ measurement frame, for which $n$ is out of [16, 8, 4, 2, 1].

The performance of the reservoir strongly relies on the dynamics of the skyrmion. Increasing skyrmion dynamics reduces reservoir operation timescales, thereby necessitating faster input application and readout speeds. Faster dynamics reduces the effect of pinning, therefore reducing the current to move the skyrmion, but also increases Brownian motion[5]. However, at increased diffusion, these random motions diminish performance, therefore faster inputs are necessary to mitigate the impact of diffusive Brownian motion. Skyrmion dynamics can be tuned by various means. Firstly, temperature can be manipulated; higher temperatures enhance dynamics[5]. However, this also decreases skyrmion size in the used samples[49], impairing readout via MOKE and making this an unfavorable method. Another method involves using the out-of-plane (OOP) DC magnetic field to modulate dynamics. We find that similar to temperature adjustments,

this can pose limitations on Kerr microscopy readout as the changes in dynamics are a result of changes in skyrmion size and small skyrmions are harder to detect for the optical detection used here[49]. The final option is employing an oscillating OOP magnetic field, which minimally affects readout while significantly boosting dynamics by orders of magnitude as previously shown[24].

To demonstrate the device's capability, the input comprises gesture data, which is processed to distinguish various human hand gestures, encoded as class numbers (see Fig. 3a). The Doppler radar-captured data is fed into the skyrmion reservoir, as elaborated in the methods section, such that each distinct hand gesture corresponds to a unique 23 frames long time-dependent voltage signal. An example can be seen in Fig. 1. The experimentally used subset of the dataset of 4800 recorded gestures employed in this study comprises four distinct hand gestures, with each gesture being repeated 50 times, thereby yielding a cumulative total of 200 recorded instances. While the large skyrmions and their dynamics used here are motivated by the limited resolution of the Kerr microscopy read-out, the use of MTJs for the read-out will allow for scaling of both the spatial device size as well as the dynamic timescale over a broad range to detect many types of signals.

To find the best position of the readout MTJ on the device, every possible position of the MTJ is evaluated independently. The emulated

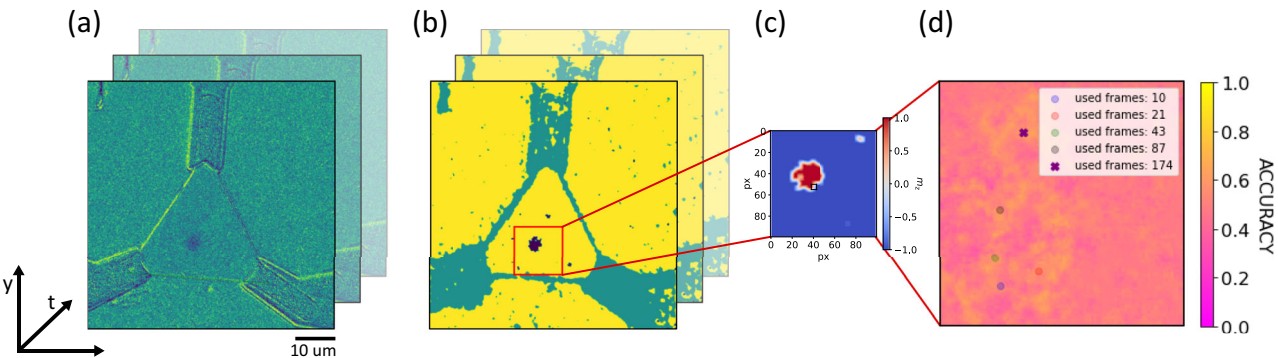

**Fig. 2 | Data processing. a** Kerr-microscope frames of the device with a skyrmion inside. **b** Convolutional Neural network prediction to reduce noise. Skyrmion prediction in purple, contacts and defects in turquoise, background and magnetic material in yellow. The skyrmion prediction is used to simulate the expected output signal from the artificial MTJs. **c** Analyzed video part with one possible emulated MTJ marked as a black square. **d** Validation accuracy map of SVM for different input pixels. In this case for differentiating all gestures with 87 data points in time. Locations with the highest accuracy for different number of data points in time are marked. As the emulated MTJs have a diameter of $0.5\,\mu m$, only the center is marked.

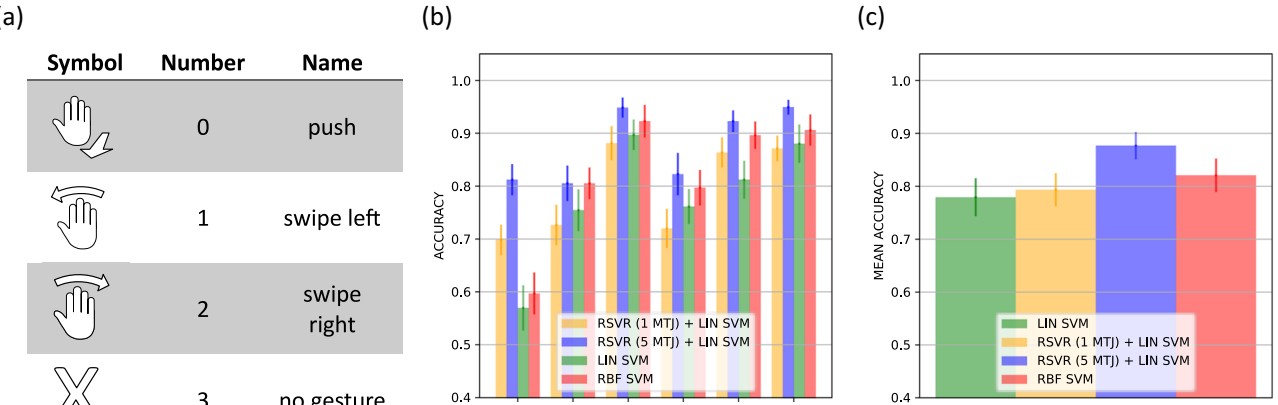

**Fig. 3 | Performance comparison. a** Table of different hand gestures in data set. **b** Comparison of detection accuracy between linear support vector machine (LIN SVM), non-linear radial basis support vector machine (RBF SVM) and reservoir (RSVR), with different number of magnetic tunnel junctions (MTJ), plus SVM (LIN SVM) for different gesture pairs. The error is calculated with the standard error of different SVM resulting from the K-Folding. **c** Mean accuracy for the different approaches, the error is the mean of the standard error from **b**).

MTJs always correspond to multiple pixels that match their size. We train a linear support vector machine (SVM) on the signal of the MTJ for each position to see which poistion performs the best to differentiate the different gestures. Note that this optimization must only be performed once. The optimization procedure produces an accuracy map of the device, see Fig. 2, which also illustrates all previous steps. Each time a state-of-the-art linear SVM is trained while K-Folding with $K = 10$ is used for robust evaluation of the validation set performance. Further details on the training of the readout layer can be found in the methods section.

Figure 3 shows the benchmark comparison for our reservoir compared to the state-of-the-art software detections. Comparing the detection accuracy of different gestures, we find a very competitive performance of the reservoir using only one MTJ for the readout. Especially for gesture pairs 01 and 13, we find even an improvement of the skyrmion reservoir compared to a linear SVM (LIN SVM). Here, the best performing number of data points were chosen for the reservoir accuracy (see methods section for more information). When using multiple MTJs for the readout, the detection accuracy increases in all cases, resulting in performance that is competitive with, or even superior to, the energy-intensive non-linear radial basis function SVM (RBF SVM). Using 5 MTJs for the readout results in faster training with improved performance and lower energy consumption compared to the RBF SVM (see Supplementary Material for more details). Figure 4 illustrates the added complexity of the RBF SVM in terms of inference time per gesture, which is the time required to perform the computation for the readout layer for one gesture. Since the inference time is directly proportional to energy consumption, this demonstrates a significant improvement in energy efficiency when using the reservoir readout compared to the RBF SVM (see Supplementary Material for more details). The decrease in inference time with larger dataset sizes can be explained by the "warm-up" time, which becomes negligible at larger dataset sizes.

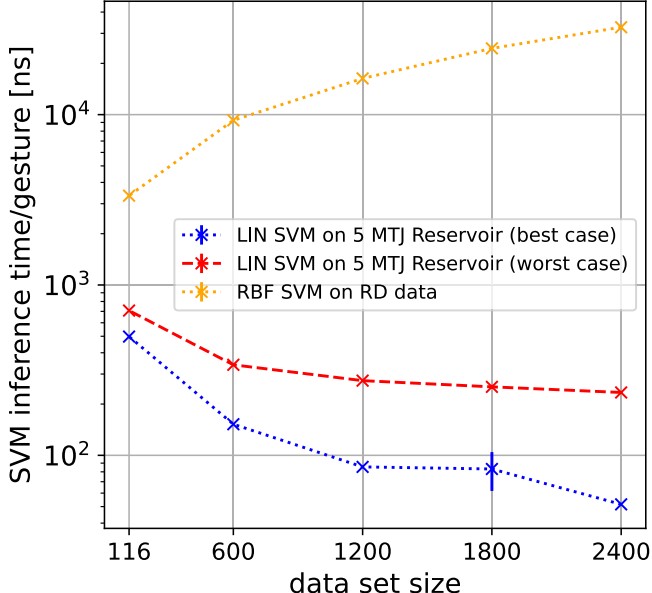

**Fig. 4 | Inference time.** Comparison of the inference time per gesture for the linear readout layer of the reservoir (LIN SVM on 5 MTJ Reservoir) versus a non-linear radial basis support vector machine (RBF SVM) operating directly on the Range-Doppler (RD) data. The best- and worst-case scenarios demonstrate inference times based on the number of Kerr images used. In the worst-case scenario, all Kerr frames are used, resulting in a more complex readout layer. Each benchmark is run 1000 times for each data point.

## Discussion

We successfully demonstrated that our skyrmion reservoir can be used as a time-multiplexed reservoir computer for time series classification problems, exemplified on Range-Doppler gesture data. In the pairwise classification, the reservoir is competitive and can even exhibit superior performance compared to state-of-the-art software-based classification methods. We note that while the performance with a single MTJ readout is already competitive with a linear classifier, using multiple MTJs as readouts can easily improve performance. This approach captures more regions of the skyrmion trajectories, resulting in performance that rivals complex, energy-intensive non-linear classifiers.

We note that a key advantage of our approach is that the device can be adapted to other tasks easily. The complexity and recognition accuracy can be optimized using a different device layout or employing multiple skyrmions. Another way to increase complexity is by connecting the third contact to another time dependent voltage, which could be explored in a future study. Furthermore, the timescale of the thermal dynamics that governs the timescale of the detected signals can be tuned by a small AC excitation over multiple orders of magnitude[24].

At the current densities used in this work, skyrmions consistently remain in the creep regime. Therefore, increasing the currents will facilitate faster motion, resulting in enhanced detection accuracy. A future step involves enhancing edge repulsion to allow for larger currents without annihilating the skyrmion. This can be accomplished by modifying the local anisotropy through focused ion beam irradiation at the corners and this will be investigated in the future.

When using a genuine MTJ instead of an emulated counterpart, it eliminates the need for optical readout and the Convolutional Neural network[50] prediction (see methods section), thereby ensuring CMOS compatibility both at the input and output stages of the reservoir. With no optical readout required, the device can be miniaturized to the nanoscale, enhancing energy efficiency and operational speed, as detailed in the Supplementary Material. The rationale behind this lies in the reduction of the skyrmions' displacement time and current-induced motion's dependence on current density, both favoring smaller devices, which thus, need less energy while additionally allowing for a larger device density. Coupled with the reduced displacement time of skyrmions within smaller devices, this scaling could be used as a variable timescale and ultra-low energy reservoir computing implementation, offering promising prospects for future applications.

In comparison to other implementations of reservoir computing using skyrmions[22,37,51], our approach stands out as the first to use complex real-life data while incorporating a scalable, highly CMOS-compatible design operating at room temperature. As a key feature, it allows for the adjustment of timescales to address a variety of different problems occurring at different timescales.

## Methods
### Sample preparation

The multilayer stack employed in this study consists of the following layers $Ta(5)/Co_{20}Fe_{60}B_{20}(0.95)/Ta(0.09)/MgO(2)/Ta(5)$, where the values enclosed in parentheses indicate the respective thickness in nanometers, and the subscripted numbers represent the concentration in percentage. The stack is deposited using a Singulus Rotaris magnetron sputtering system. The sample exhibits perpendicular magnetic anisotropy (PMA) due to the interface between the MgO and the ferromagnetic layer $Co_{20}Fe_{60}B_{20}$. The bottom Ta layer is used to induce the spin-orbit torques into the ferromagnetic layer. While the Ta (0.09) dusting layer between MgO and $Co_{20}Fe_{60}B_{20}$ adjusts the strength of the PMA and generates a low-energy landscape for magnetic skyrmions.

The 5 nm thick capping layer consisting of tantalum prevents oxidation, together resulting in a multilayer stack that hosts skyrmions at above room temperature experiencing low pinning and thermal diffusion. The skyrmion diameter is roughly $4-5\,\mu m$ depending on temperature and the OOP magnetic field amplitude.

The Raith Electron Beam Pioneer system was employed for electron beam lithography in order to fabricate patterned structures. The sample underwent etching with Argon ions through the IonSys Model 500 ion beam etching system.

The device consists of an equilateral triangle with an edge length of $36\,\mu m$. At the corners extra rectangles are attached with a width of $4\,\mu m$ and length of $15\,\mu m$ for better contact connectivity. These contacts are composed of a 5 nm layer of chromium overlaid with 60 nm gold and were fabricated utilizing electron beam lithography in conjunction with a lift-off technique. The device under examination exhibits a measured resistance of 1.1 kOhm between the two contacts.

## Measurement setup

To visualize the skyrmions, the experimental setup employs a microscope manufactured by evico magnetics GmbH, which leverages the magneto-optical Kerr effect (MOKE). The microscope is connected to a CCD camera recording at 16 frames per second with an exposure time of 62.5 ms and a resolution of $1344 \times 1024$ pixels. With a 2x2 binning, the resolution is reduced to $672 \times 512$ pixels in favor of a higher signal-to-noise ratio. The camera captures an area of $80 \times 61\,\mu m^2$. To achieve better contrast a differential image between the skyrmion state and the saturated state is used.

The setup consists of custom-designed coils, featuring one coil for generating the out-of-plane (OOP) magnetic field and an additional pair for the in-plane (IP) field. To the OOP coil, a Peltier element is attached to change the temperature between range from $285-360$ K, with a thermal stability of 0.3 K. In the used setup, a temperature of 325 K was chosen, measured by using an PT100 resistive sensor. Furthermore, to enhance the thermal stability, the entire microscope system is housed within a laminar flow enclosure equipped with precise temperature control. This strategic implementation results in minimal thermal drift within the experimental setup, facilitating extended measurement durations at higher magnification levels.

The skyrmion is nucleated with an IP field pulse of 20 mT and a stabilizing OOP bias field of $100\,\mu T$. After nucleation, the additional OOP fields oscillations are started with an amplitude of $50\,\mu T$ and a frequency of 100 Hz, while keeping a bias of $100\,\mu T$. As the input signal, a time dependent voltage is applied to two corners using an arbitrary waveform generator. The recorded imagery is synchronized to the waveform generator.

## Data evaluation

The dataset consists of 4800 measurements of 4 classes of gestures: "0: push", "1: swipe left", "2: swipe right" and "3: no gesture" and was already successfully used to train spiking neural networks[52]. Varying

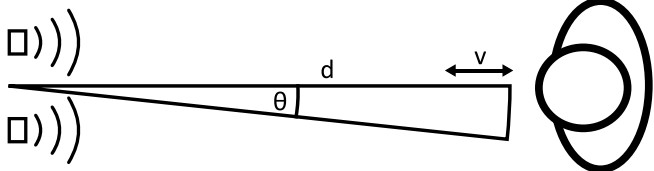

**Fig. 5 | Radar setup.** Top view schematic representation of the radar setup utilized in the dataset. On the left are the two radar sensors, while on the right is the individual performing the hand gestures. Here, $d$, $v$ and $\theta$ represent distance, velocity and angle respectively.

people stood at the same position in regard to a radar sensor and performed hand gestures according to the classes. Every gesture measurement consists of 23 frames taken in real-time. The motion was detected by two Infineon Technologies radars sensors of the type BGT60TR13C. The setup can be seen in Fig. 5. The used sensor is a frequency-modulated continuous wave radar employing a chirp-sequence of 64 chirps per frame, with 32 sample points per chirp[52]. The data has to be transformed by exploiting micro Doppler effects[53], and obtaining Doppler maps, as described in ref. 54. and ref. 55. Mainly, Fourier-transformations are performed to obtain the Doppler frequency shifts, resulting in two-channel maps (Range-Doppler and Range-Angle), showing the angle or the amplitude against the relative velocity and the distance to the radars. Here the signal of a single sensor is sufficient to result in the Range-Doppler maps, while the phase difference to the second sensor is utilized for the Range-Angle maps. Figure 6 shows one exemplary "0 – push" gesture after the Range-Doppler (RD) and the Range-Angle (RA) transformation.

As we want to minimize the data fed into the reservoir, we restrict ourselves to only one voxel of the maps. To find the one containing the most information, we perform multi-class radial basis functions Support Vector Machine (RBF SVM)[56,57] classification on every possible index (Range-Doppler/Range-Angle), to find the input voxel (23 consecutive time-steps of one particular index) that performs best on all four gestures at once. We further checked if data pooling (Average of $1 \times 1$, $2 \times 2$, or $4 \times 4$ tiling) increases classification. Every input was trained with K-Folding and $K = 10$ to obtain statistically robust values. This means there are always ten RBF SVMs trained, each time using a different one-tenth of the data as the test set. The best score was obtained by averaging the performance of the $K$ test sets. The respective voxel is marked with a red box in Fig. 6. Resulting exemplary signals can be found in Fig. 7. Reducing the data to one input is also required to obtain experimental data in a reasonable time, as the manual experiments are time-consuming and must be taken with care. In a non-proof-of-concept device, this training process could of course be automatized using specifically tailored hardware. The pairwise results shown in the main text are obtained by the same voxel, namely the one that performed best.

After the input data was defined, signal-and-time matching was necessary to transform the arbitrary input signal into a current density that significantly moves the skyrmion without annihilating it. In addition, the time scales of the input need to match, e.g., the input signal should vary at similar timescales as the skyrmion dynamics happens. The matching functions read as:

$$U(t) = c_1 \cdot \tanh((v(t) - \mu) \cdot c_2) \qquad (1)$$

With $v(t)$ being the signal at a specific time frame $t$, $\mu$ the mean value of the dataset (to normalize them around zero. The other parameters were chosen with $c_1 = 3.5$ mV and $c_2 = 4$ in such a way that the skyrmions are driven far enough out of equilibrium to distinguish trajectories, but not too much that there is no significant probability for a skyrmion to be annihilated at the boundary. The 23 timeframes were mapped to an input. After each measurement, we let the skyrmion relax back to the center position, which depending on the diffusion coefficient takes milliseconds to seconds. Also, the gesture order was randomized to reduce the possibility of any bias regarding potential insufficient relaxation.

As typically some threshold current strength needs to be overcome to move the skyrmion at all, we also use the tanh function to effectively increase the normalized input. To validate that the nonlinear transformations do not reduce the complexity of the problem, the reference pure software-based approaches were also trained using

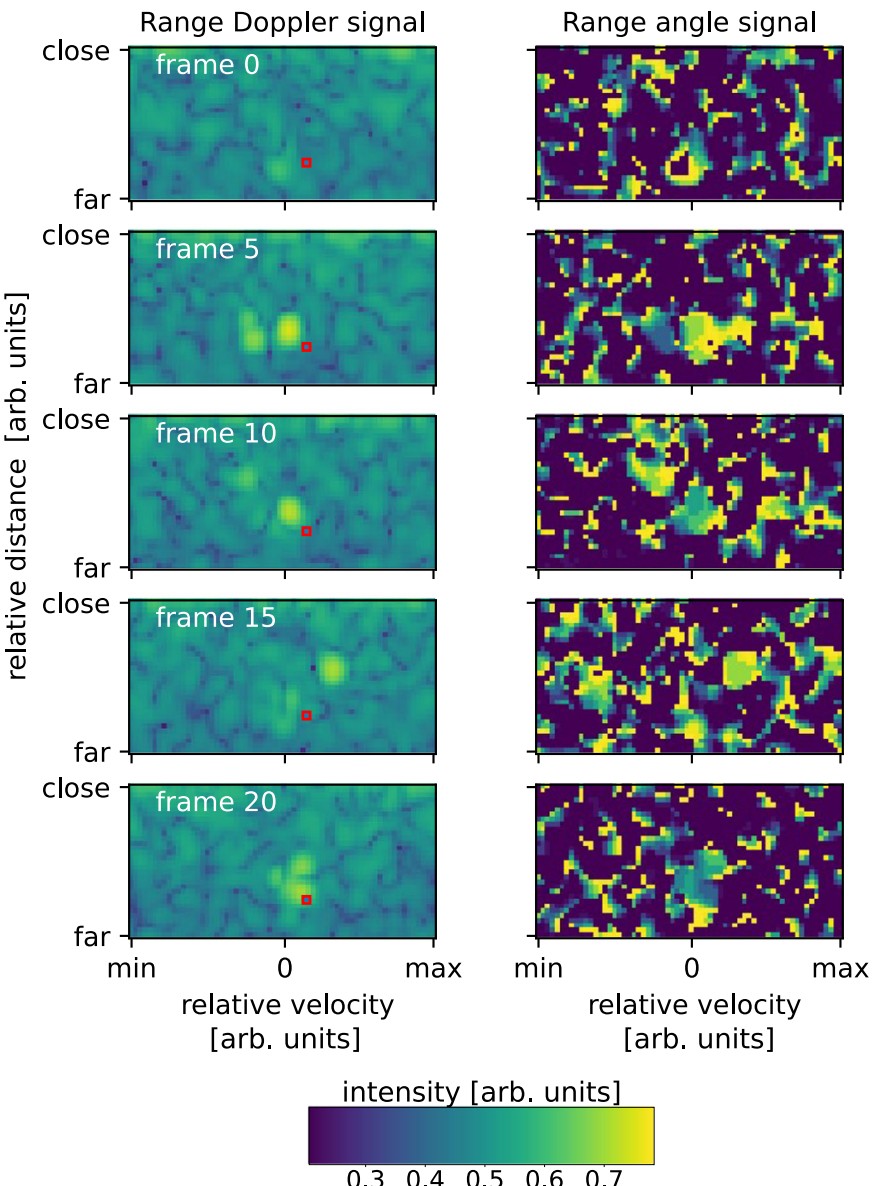

**Fig. 6 | Range-Doppler and Range-Angle maps.** Range-Doppler maps on the left and Range-Angle maps on the right for one specific gesture and one-time frame. With the Range-Angle maps the angle is given by the amplitude. Apart from noise, one can detect two blobs in the maps, one referring to the person's body gesticulating, and the other to the hand doing the movement. We also marked the voxel with a red box that was used as input for our skyrmion reservoir.

the time dependent voltage signal fed into the reservoir. Here, no significant differences were found.

After the MOKE measurements were performed, the data was cropped to 174 timeframes per gesture. The contrast was then enhanced using the 2.5% outer percentiles for adjustment, after setting extreme greyscale values (that might occur due to the combination of background-subtraction and sample-drift) to the mean value of each image. Then, to detect the skyrmion, a Convolutional Neural network was used which can reliably segment the image into the skyrmion (label: 0), defect (1), and magnetic background (label: 2)[50], while for our analysis we reduce the labels to (Skyrmion: +1 and Background/defects: −1). Effectively this converts the MOKE contrast into a normalized 2D out-of-plane magnetization map of the sample.

To mimic the MTJ-readout, we chose a rectangular size of 5 × 5 pixels, while the MOKE resolution is ≈120 nm/pixel. An MTJ of this size can easily be manufactured[58]. We average the segmented label in that area to obtain more than two discretizations

of the output space and interpret the value as a normalized resistance of the MTJ (0-lowest resistance, −1 highest resistance or vice versa). The readout layer of the reservoir computer consists of a SVM that is fed with the time dependent signal of the MTJ. We train the readout SVM independently on every possible position (MTJ is shifted by 1 pixel in each direction) in the confinement and obtain accuracy maps for the reservoir (Fig. 2d). We also check how many time frames of the MTJ signal are optimal, while we scan a range of every $n$th timeframe, with $n \in [16, 8, 4, 2, 1]$ and again K-Folding with $K = 10$, while the benchmarking was chosen to be the maximum value of the average accuracies of the test sets. Figure 8 visualizes the mean performance of training and test set for different number of used Kerr microscopy frames. We compare the outputs of these SVMs in the main text, Fig. 3.

For adding a 2nd MTJ we include the best-performing input data of the single MTJ results and scan every possible MTJ position again, adding the information of the 1st MTJ and the data

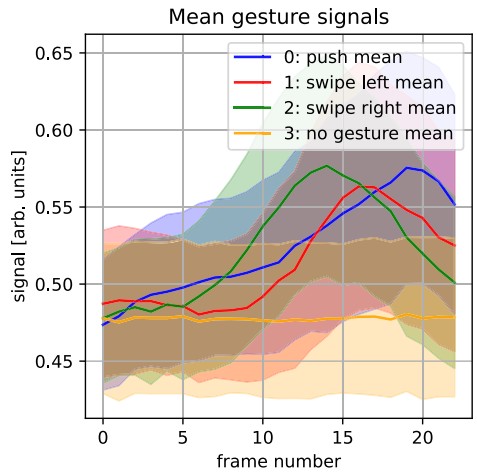
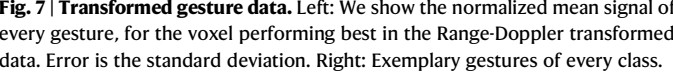
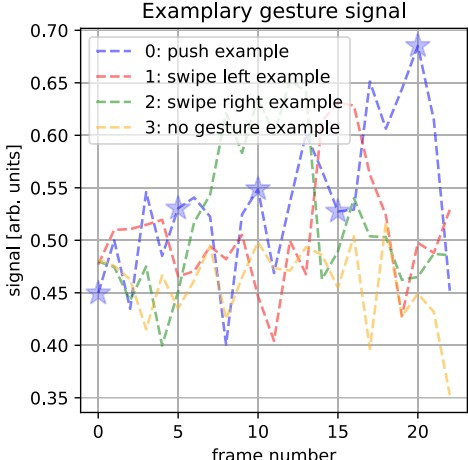

**Fig. 7 | Transformed gesture data.** Left: We show the normalized mean signal of every gesture, for the voxel performing best in the Range-Doppler transformed data. Error is the standard deviation. Right: Exemplary gestures of every class. The blue star marks the values which are shown in red boxes in Fig. 6, which shows the same gesture.

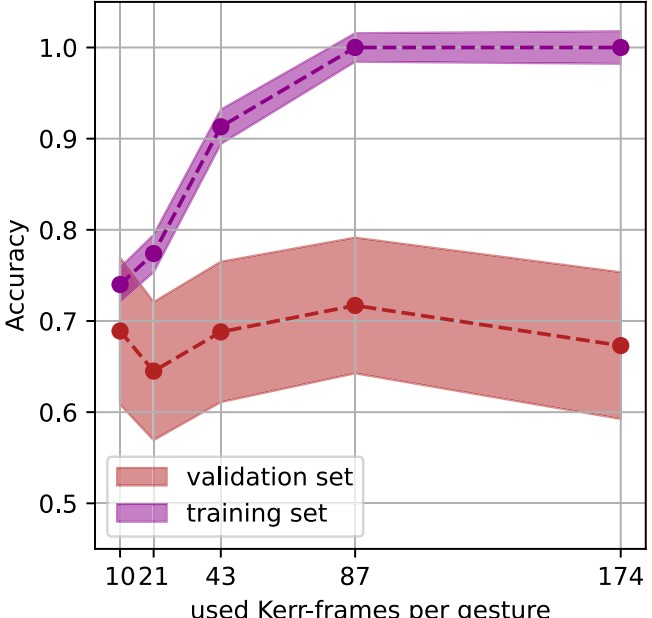

**Fig. 8 | Performance of the Reservoir, when different amount of Kerr video frames is used for the evaluation.** The plots show the pairwise gesture comparison 0-push vs 1-swipe left. Error is the standard error. We see that the performance on the validation set peaks at specific amount of input (in this case 0.708 for 87 data points), while above we enter an overfitting regime of the linear SVM. For the plots in Fig. 3 in the main text, we plot the best performance we find in the validation set. The best MTJ position for varying input size is also indicated in Fig. 2d) in the main text.

from the current scan. The process can be repeated for further additional MTJs.

## Data availability
The data supporting the findings of this work are available from the corresponding authors upon request.

## Code availability
The computer codes used for data analysis are available upon request from the corresponding author.

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

## Acknowledgements

The gesture data were kindly provided by Infineon for use in this research. The work in Mainz (G.B., T.B.W., K.R., M.A.B., F.K. and M.K.) was supported by the Deutsche Forschungsgemeinschaft (DFG, German Research Foundation) projects 403502522 (SPP 2137 Skyrmionics), 49741853, and 268565370 (SFB TRR173 projects A01, B02 and A12) as well as TopDyn and the Zeiss foundation through the Center for Emergent Algorithmic Intelligence. This research was supported by the National Research Council of Science & Technology (NST) grant by the Korean government MSIT (Grant No. GTL24041-000). J.H.M. acknowledges funding VIDI talent programme of the Dutch Research Council (NWO), project 223.157 (CHASEMAG). The work is a highly interactive collaboration supported by the Horizon 2020 Framework Program of the European Commission under FET-Open grant agreement no. 863155 (s-Nebula) and ERC-2019-SyG no. 856538 (3D MAGiC) and the Horizon Europe project no. 101070290 (NIMFEIA), which M.K. and J.H.M. acknowledge. M.A.B. thanks the DFG TRR146 for partial financial support. M.A.B. is supported by a doctoral scholarship from the Studienstiftung des deutschen Volkes. We would additionally like to

acknowledge helpful discussions with Peter Virnau. M.K. thanks the Norwegian Research Council for support through its Center of Excellence 262633 "QSpin".

## Author contributions

G.B. performed the experiments and wrote the manuscript with T.B.W., K.R., M.A.B., S.K., J.H.M. and M.K. T.B.W. performed the analysis and M.A.B. helped with the analysis. K.R. helped prepare the samples while F.K. grew the materials stacks used in the experiments. P.G. and K.K. provided the gesture data. J.H.M. and M.K. supervised the project. All authors commented on the final manuscript.

## Funding

## Competing interests

The authors declare no competing interests.
