## [Peer Review File · Nature Communications]

REVIEWER COMMENTS

Reviewer #1 (Remarks to the Author):

In this work, the authors G. Beneke et al. take a device based on a triangular multilayer consisting of a single skyrmion [same as that proposed in some of the same authors' previous paper Nat. Comm. 13, 6982 (2022)] to perform its application to the hand gesture recognition task. While their above paper showed the ability of a single skyrmion in this system to separate the nonlinear logic gates, this manuscript aims to further consolidate the potential of the same device for practical applications with low power consumption. In my opinion on authors' previous work, their design of this device very cleverly utilizes the confined geometry to enhance the nonlinearity of the skyrmion's position with respect to input voltages, and the auto-relaxation of the skyrmion to the center position after input injection is a further advantage that guarantees repeatable computational results. On the other hand, physical reservoir computing is currently a hot topic since it greatly reduces the issues of unstable and/or time-consuming trainings in neural networks. Together with the room-temperature plus ultra-low-current operation of this nanometric device by the authors, there is no doubt this manuscript will arouse a wide interest of general readers. However, while their motivations are well-stated and experiments are well done and reliable, I found several insufficiencies in the current version of this manuscript that either cause confusion or render it not informative enough to be a new publication in Nature Communications, as listed in questions below. Although some of them may stem from my own misunderstandings due to limited knowledge, I would recommend the publishing of this manuscript in Nature Communications only after all of them are clarified and/or complemented correspondingly in the manuscript by authors.

1. (Major question) The authors' major result is Figure 3(b). While they claim that the skyrmion reservoir can lead to competitive ability compared to SVM for gesture recognition, I found this claim may not be quite convincing due to following sub-questions.

(1a) Why do the authors compare "SVM" versus "SVM + reservoir", rather than compare "SVM" versus "reservoir only"? Is it not possible to perform the task by the reservoir without SVM? It seems the SVM is required to find the best position of readout MTJ for entering the input into reservoir. After this procedure, the response of skyrmion reservoir only leads to comparable results compared to using SVM only, then why don't we use SVM directly, without the reservoir?

(1b) Corresponding to (1a), for the case of SVM only, what is the entire procedure? Is it more demanding in any aspect than using SVM + reservoir? To me, it is not transparent to see the benefit of adding the reservoir, since the comparison of these two procedures are not quite clearly described. This is related to question 2 below.

(1c) When the reservoir is added, the error bar is enhanced compared to using SVM only. What causes this behavior? Is it possible to reduce this error? More importantly, this enhanced error seems to invalidate authors' claim of competitive ability by skyrmion reservoir, since adding reservoir to SVM only leads to comparable averaged recognition accuracies but much severer standard deviations from different folds (testing sets). For practical applications, we desire a device to be stable for each operation, but the large errors of around +/-20% accuracy seem not to guarantee good stability of the device. Are there proposals by authors to improve it?

2. (Major question) There are some ambiguities about the training procedure in this work as follows.

(2a) What exactly is the training procedure in this skyrmion device? In the common reservoir computing setup, we define input, output (mostly a linear combination of measured responses in reservoir), and some loss function to be minimized by training. For training process there are also various methods like gradient descent iteration or pseudoinverse matrix. However, in the manuscript, except for input, the definitions of following items seem to be described unclearly or in nowhere: (i) reservoir state (I would expect the reservoir state as formed by skyrmion's position vectors $(x(t), y(t))$ in time (or frame number), is it correct?), (ii) output function, (iii) loss function, (iv) number of training/testing datasets (Is the number of testing set 10 as the 10 folds? Then what about training set, and equivalently how many different gestures are used for training/testing?), (v) dimension (number of virtual nodes) of weight matrix required for optimization, and (vi) the training algorithm.

(2b) Is it comparable or even faster to train the skyrmion reservoir than the SVM model, such that the reservoir can gain some computational speed as an advantage?

(2c) In page 9, it is mentioned an additional convolutional neural network is used to detect skyrmion. Is this additional setup demanding in computation that will decrease the power of the reservoir? I ask this since naïvely one expects that the reservoir computing setup in any physical system should be a framework which no longer requires the trainings for hidden layers in neural networks. Perhaps I got some misunderstandings in these questions, but hopefully authors could clarify them.

3. (Major question) The current manuscript only takes a specific set of physical parameters (e.g., temperature, amplitude (and frequency) of OOP dc (ac) fields, input voltage amplitude, etc.) to do the gesture recognition. Figure 3, after possible modifications according to above two major questions, although may be a sound demonstration of the ability of this skyrmion reservoir, the overall manuscript to me still lacks sufficient comparisons and surveys of different parameter

regimes that can make it a more informative publication in Nature Communications. I separate this into sub-questions below.

(3a) Temperature: The temperature-induced Brownian motion of skyrmion is beneficial to overcome the pinning as mentioned by authors. Then, in the depinning regime, what is the effect of temperature? I would expect higher temperatures cause more randomness or noises of skyrmion motion thus decrease the recognition accuracy, is it correct empirically?

(3b) Amplitude of OOP dc field may change the size of skyrmions. Is there any effect of skyrmion size on its recognition ability?

(3c) Amplitude/frequency of OOP ac field: In the beginning part of the manuscript, pages 2 and 3, authors already emphasize on the advantage of tuning the diffusion timescales by the OOP ac field to avoid energy-consuming rescaling of the dynamical timescales of hand gestures. However, in the main result there is no evidence at least to qualitatively support their claim of this advantage via, e.g., comparing recognition results using different amplitudes/frequencies of OOP ac field while fixing the same time rescaling. I think future readers will expect to see this kind of comparison after they read those statements in pages 2 and 3, and it will give a coherent and convincing proof of authors' claim.

(3d) Input voltage amplitude: Authors have clearly mentioned the skyrmion might be pinned in middle or annihilated at contact regions if the input voltage is too small or too large, respectively. Then in the voltage range without skyrmion pinning and annihilation, is there correlation between recognition accuracy and input voltage empirically?

Although new experiments are not mandatory since they may require long times for authors to revise this work, for all or some of the parameters listed above, could authors at least provide some qualitative argument or description about their effects on the recognition results?

4. (Minor question) Some minor sub-questions are in order.

(4a) While two of the contacts are applied by the input voltage difference, there seems no description about the third contact. Is it grounded?

(4b) In page 5, the third paragraph, the sentence "It is also visible that the non-linear transformation to improve the skyrmion response does not change the accuracy significantly for the pure software-based approach." is not clear. What does it mean? What is the non-linear transformation?

5. (Minor question) In the Results section and Methods section, authors describe their material. Following information may be better to add for future readers. (i) In which (or both?) of the ferromagnetic CoFeB or MgO layers that skyrmion lies? (ii) What is the typical skyrmion size in this system (for the specifically applied OOP dc field)? (iii) How does this system induce spin-orbit torque by current? I suggest adding information of these even though they must be trivial to experts in this field.

6. (Minor question) In the section of Data Evaluation, the procedure of gathering the dataset may be described in a clearer way. I suggest some modifications as follows for better understandings by future readers.

(i) For instance, a simple schematic figure showing the person's hand doing gesture together with the radar system may be good, since to me it is not clear that, in the sentence "... showing the angle or the amplitude against the relative velocity and the distance to the radars", what the angle and amplitude actually mean? In this schematic figure the distance/angle could be clearly indicated.

(ii) If possible, perhaps authors could provide more details in this figure and/or in text of what data they measure in these experiments to do the Fourier transform.

(iii) In Fig. SUP1, the color bars should be shown.

7. (Minor question) Some typos are as follows:

(i) In page 3, the 7th row of the last paragraph, the sentence is a bit confusing, "... is realized by..." may need to be modified as "... as realized by..."

(ii) In caption of Fig. 2, 1st row, "... a with..." should be modified to "... with a..."

(iii) In caption of Fig. 3, "... b) Comparison detection..." might need to be modified as "... b) Comparison of detection...". Also, it is better to add an abbreviation such as "... reservoir (RSVR)..." in this caption.

Reviewer #2 (Remarks to the Author):

In this manuscript, the authors present gesture recognition using reservoir computing powered by the current-induced dynamics of skyrmions. In particular, the authors claim that when the time scale of input signals is close to that of skyrmion dynamics, the performance of the gesture recognition task is comparable to that of state-of-the-art software-based neural networks, without the need for preprocessing of data.

The manuscript is clearly written and refers to previous literature appropriately. In addition, the quality of the data and presentation are satisfactory. However, I believe this paper does not meet the high standards of Nature Communications for the following reasons.

1.The first point concerns the significance of this paper. So far, experiments on reservoir computing using skyrmion dynamics have been reported by several groups [for example, K. Raab, et al., Nat. Commun. 13, 6982 (2022), T. Yokouchi et al., Science Advances 8, abq5652 (2022), and O. Lee et al., Nat. Mat. 23, 79 (2023)] (one is the author's group). While the detailed structure differs in each paper, the key concept of applying skyrmion dynamics to physical reservoir computing has already been established by these works. Hence, the main significance of this paper lies in the demonstration of physical reservoir computing without the need for preprocessing of data. However, it seems obvious that preprocessing is not required when the time scale of input data matches that of the reservoir system. Of course, proving the idea experimentally is quite important. However, it is questionable whether this paper will attract much attention from general audiences in a broad field, and thus a more specific journal might be more appropriate.

2.The second point is about the conclusion of this work. The authors conclude that skyrmion reservoir computing is competitive and can even exhibit superior performance compared to state-of-the-art software-based neural networks, a result shown in Fig. 3. However, the error bars for the skyrmion-based reservoir are quite larger than those for the state-of-the-art software-based neural network (Fig. 3). This indicates that the performance of the skyrmion reservoir depends on the choice of data set for K-fold training. Does this imply that the performance of the skyrmion-based reservoir is less than that of the state-of-the-art software-based neural networks?

In addition, several error bars exceed the accuracy of 100 %. What does an accuracy above 100% mean? It would be better to show all data points instead of presenting the average value and errors or to reconsider the calculation method of error bars.

REVIEWER COMMENTS

Reviewer #1 (Remarks to the Author):

In this work, the authors G. Beneke et al. take a device based on a triangular multilayer consisting of a single skyrmion [same as that proposed in some of the same authors' previous paper Nat. Comm. 13, 6982 (2022)] to perform its application to the hand gesture recognition task. While their above paper showed the ability of a single skyrmion in this system to separate the nonlinear logic gates, this manuscript aims to further consolidate the potential of the same device for practical applications with low power consumption. In my opinion on authors' previous work, their design of this device very cleverly utilizes the confined geometry to enhance the nonlinearity of the skyrmion's position with respect to input voltages, and the auto-relaxation of the skyrmion to the center position after input injection is a further advantage that guarantees repeatable computational results. On the other hand, physical reservoir computing is currently a hot topic since it greatly reduces the issues of unstable and/or time-consuming trainings in neural networks. Together with the room-temperature plus ultra-low-current operation of this nanometric device by the authors, there is no doubt this manuscript will arouse a wide interest of general readers. However, while their motivations are well-stated and experiments are well done and reliable, I found several insufficiencies in the current version of this manuscript that either cause confusion or render it not informative enough to be a new publication in Nature Communications, as listed in questions below. Although some of them may stem from my own misunderstandings due to limited knowledge, I would recommend the publishing of this manuscript in Nature Communications only after all of them are clarified and/or complemented correspondingly in the manuscript by authors.

We are thankful to Referee #1 for his/her favorable feedback on the quality of work in our manuscript. Below, we address in-depth the points raised by the referee. We fully align the revised manuscript with the suggestions provided and ensure that the current version fulfills the publication requirements suggested by the referee. In particular, we answer the question about the advance of our approach over incumbent approaches in a detailed new study that we have added to our work. Detailed responses follow.

1. (Major question) The authors' major result is Figure 3(b). While they claim that the skyrmion reservoir can lead to competitive ability compared to SVM for gesture recognition, I found this claim may not be quite convincing due to following sub-questions.

(1a) Why do the authors compare "SVM" versus "SVM + reservoir", rather than compare "SVM" versus "reservoir only"? Is it not possible to perform the task by the reservoir without SVM? It seems the SVM is required to find the best position of readout MTJ for entering the input into reservoir. After this procedure, the response of skyrmion reservoir only leads to comparable results compared to using SVM only, then why don't we use SVM directly, without the reservoir?

We thank the referee for pointing out that we have not been sufficiently clear on this point. In the revised manuscript we now clarify this by adding a more exhaustive analysis and the resulting additional results: As previously done, we feed the doppler radar data into the reservoir and then we learn a SVM machine on each possible position of the simulated magnetic tunnel junction. This simple *linear* SVM is the output layer of the reservoir, which transforms the signal of the magnetic tunnel junction into which gesture type is fed into the reservoir (we call this RSVR + LIN SVM). We then compare our reservoir with a linear SVM to just a SVM trained on the raw data from the doppler radar. For this SVM without the reservoir, we can use a linear SVM (called LIN SVM) or a more complex radial basis function support vector machine (called RBF SVM).

As a major change compared to the previous manuscript, we have now carried out an additional study that correctly compares the performance of the reservoir with the linear SVM (RSRV + LIN SVM) and the linear SVM on the full data (LIN SVM). While previously we did use a larger data set for (LIN SVM) than for (RSRV + LIN SVM) because of the longer time it takes to carry out the experiments for the reservoir case, we now use identical data sets for both for a robust comparison. Using the same data sets and calculating the errors correctly we provide the following key finding: we find that (i) the reservoir with a single MTJ and with a linear SVM (RSRV (1 MTJ) + LIN SVM) outperforms the linear SVM (LIN SVM) and (ii) that by adding more MTJs we can significantly improve the performance of the reservoir so that a reservoir with e.g. 5 MTJs and a linear SVM (RSRV (5 MTJ) + LIN SVM)) outperforms even the advanced and energetically much more costly radial basis function SVM (RBF SVM) with the difference being clearly outside the error bars.

We show all the results in detail now in the figure that shows the accuracy for the different cases and we discuss the key findings. Firstly one can see that even the simple reservoir with only one skyrmion inside in the simplest geometry possible, while only having one small MTJ at a single location (RSRV (1 MTJ) + LIN SVM), is already able to achieve competitive performance compared to the linear SVM (LIN SVM). Furthermore, introducing multiple skyrmions, other confinements or adding more MTJs for the readout layer is a straightforward way to increase the complexity of the dynamics and therefore the performance of the reservoir.

To demonstrate this point, we have carried out significant additional work and added a new set of data using multiple MTJs for the readout (RSRV (5 MTJ) + LIN SVM), which increases the performance in all tasks and clearly outperforms the linear SVM and showing a comparable or even improved performance to a non-linear radial basis SVM (RBF SVM), which is more complex and energetically much more costly. We provide in our answer to comment 1b a more extensive comparison of the energy consumption.

The second plot compares the mean performance of the different approaches, including the mean standard error for each. This validates the improved accuracy of the reservoir with one MTJ compared to the LIN SVM. Additionally, it highlights the significantly outside the error bar improved performance of the reservoir with 5 MTJs compared to the RBF SVM.

We updated the corresponding text section and the figure in the revised manuscript for a readout with multiple MTJs.

(1b) Corresponding to (1a), for the case of SVM only, what is the entire procedure? Is it more demanding in any aspect than using SVM + reservoir? To me, it is not transparent to see the benefit of adding the reservoir, since the comparison of these two procedures are not quite clearly described. This is related to question 2 below.

In the first used case of the reservoir (RSVR (1 MTJ) + LIN SVM), the performance is at least comparable and on average better than that of a standalone linear SVM using the raw data. However, by increasing the complexity, we can improve recognition accuracy due to gaining more information on the skyrmion trajectories, thus surpassing the standalone SVM's (LIN SVM) performance.

As a first improvement, we have explored using multiple MTJs and this has improved the performance significantly. With up to 5 MTJs, the mean performance of our reservoir with a linear SVM (RSVR + LIN SVM) is not only superior to the linear SVM (LIN SVM (small set)) but even superior to a complex non-linear radial basis function SVM (RBF SVM (small set)). For the reservoir we retain the advantage of employing a simple linear readout, unlike the RBF SVM, which requires complex and energy-intensive calculations.

To validate the high energy consumption of the RBF SVM, we measure the inference time of the reservoir's readout layer (LIN SVM on 5 MTJ Reservoir) and the pure software-based approach (RBF SVM on RD data). The RBF SVM's inference time depends polynomially on the dataset size, whereas the reservoir's linear readout maintains a constant inference time. Notably, the reservoir's inference time decreases with larger dataset sizes as the "warm-up" time becomes negligible compared to the dataset size.

We also plot the best and worst-case scenarios for the reservoir readout, which varies with the MTJs' temporal resolution. This parameter is optimized for better detection accuracies and alters the dimensions of the readout layer. We observe that the reservoir readout reduces the required time by a factor of three compared to the RBF SVM with the smaller dataset used in the experiment, and this time reduction is even greater with larger dataset sizes. To ensure quantitative results, the benchmarks are run 1000 times for each data point.

Since inference time is directly proportional to energy consumption, we demonstrate a significant improvement in energy efficiency when using the reservoir readout compared to the RBF SVM. Additionally, the total energy consumption of the reservoir is smaller because the energy required to operate the reservoir is not significant compared to the readout layer. We now describe this in the revised supplementary information.

(1c) When the reservoir is added, the error bar is enhanced compared to using SVM only. What causes this behavior? Is it possible to reduce this error? More importantly, this enhanced error seems to invalidate authors' claim of competitive ability by skyrmion reservoir, since adding reservoir to SVM only leads to comparable averaged recognition accuracies but much severer standard deviations from different folds (testing sets). For practical applications, we desire a device to be stable for each operation, but the large errors of around +/-20% accuracy seem not to guarantee good stability of the device. Are there proposals by authors to improve it?

We thank the referee for this valid comment. The increased error bars can be attributed largely to the smaller dataset initially used for the reservoir learning. To allow for a better comparison, we now use the same dataset employed in the pure software-based approach (LIN SVM (small set) and RBF SVM (small set)), resulting in comparable errors. Another reason for the larger error bars in the reservoir case was an incorrect evaluation of the reservoir, leading to an overestimation of the error. After correcting both issues, the reservoir and the pure software-based approach achieve comparable errors, as expected. We also decided to show the standard error compared to the standard deviation, to leverage the statistics of the K-Folding. We apologize for the initial mistake in providing an incorrect comparison, but we now present a robust comparison in the revised manuscript that also makes our message clearer.

2. (Major question) There are some ambiguities about the training procedure in this work as follows.

(2a) What exactly is the training procedure in this skyrmion device? In the common reservoir computing setup, we define input, output (mostly a linear combination of measured responses in reservoir), and some loss function to be minimized by training. For training process there are also

various methods like gradient descent iteration or pseudoinverse matrix. However, in the manuscript, except for input, the definitions of following items seem to be described unclearly or in nowhere: (i) reservoir state (I would expect the reservoir state as formed by skyrmion's position vectors $(x(t),y(t))$ in time (or frame number), is it correct?), (ii) output function, (iii) loss function, (iv) number of training/testing datasets (Is the number of testing set 10 as the 10 folds? Then what about training set, and equivalently how many different gestures are used for training/testing?), (v) dimension (number of virtual nodes) of weight matrix required for optimization, and (vi) the training algorithm.

We thank the referee for raising this point and are glad to explain the entire training process in more detail in the revised manuscript:

(i) The full device state can be viewed as the magnetization of the sample, which encodes the skyrmion's position. When the skyrmion is in the center of the device (apart from thermal fluctuations around this equilibrium position), we consider it to be in standby mode or the starting state. When a signal is applied to the device, the skyrmion is displaced from the center. To mimic all-electrical MTJ read-out, we only consider a small area of the device that defines the readout area as the actual reservoir state. This is represented by the average magnetization within the area of an MTJ pillar, which is then proportional to the electrical MTJ signal.

(ii) For every frame considered (dependent on the sampling rate, which we used as a meta-parameter), the output function is the average magnetization of the effective MTJ read-out area.

(iii) A linear SVM training process minimizes the distance of samples to the decision boundary (which is $N-1$ dimensional). Thereby, only false-classified examples and samples with some small distance d to the boundary are used to define the loss.

(iv) Due to the K-Folding of 10, there are 10 sets of training and test sets, where each time another $1/10$ part of the hole set is the test set. The shown accuracies are always the mean over these 10 separately learned output layers, while the error in the revised manuscript is the standard error.

(v) The number of weights for a linear SVM is simply the number of dimensions (N), providing enough parameters to define a hyperplane in N -dimensional space.

(vi) Since the reservoir has no trainable weights, we only train the SVM using the standard training process provided by the scikit-learn package [Pedregosa, F. *et al.*, *Journal of Machine Learning Research* **12**, 2825–2830 (2011)].

To explain these points in more detail, we adjusted and enhanced the “Data Evaluation” section of the revised manuscript.

(2b) Is it comparable or even faster to train the skyrmion reservoir than the SVM model, such that the reservoir can gain some computational speed as an advantage?

We apologize for not making this clear. As the training was not sufficiently explained in the original manuscript, we are glad to elaborate more on the training used:

The key advantage of reservoir computing is that the reservoir itself is typically not trained. It functions as a "black box," essentially a recurrent neural network with fixed weights. Only the readout layer is trained. When comparing the reservoir readout with a linear SVM model, they perform similarly. However, when comparing the reservoir with multiple MTJs against a non-linear

radial basis function (RBF) SVM, the training of the reservoir readout layer is faster and therefore more energy-efficient.

In the revised manuscript in addition to including a comparison using multiple MTJs for the readout and contrasting it with the RBF SVM, we have also provided detailed information on the training process.

(2c) In page 9, it is mentioned an additional convolutional neural network is used to detect skyrmion. Is this additional setup demanding in computation that will decrease the power of the reservoir? I ask this since naïvely one expects that the reservoir computing setup in any physical system should be a framework which no longer requires the trainings for hidden layers in neural networks. Perhaps I got some misunderstandings in these questions, but hopefully authors could clarify them.

We are grateful for the reviewer bringing up this question, as the use of the U-Net neural network was not explained in detail in the original manuscript.

We now explain this in the revised manuscript:

In a final device, the skyrmion position would be read-out by a CMOS-compatible electrical device, e.g. by an MTJ that is stacked on top of the device at the optimal position, as obtained by this study. While MTJ read-out has been realized experimentally (see e.g. [Chen, S. *et al.*, *Nature* **627**, 522–527 (2024)]), the focus of our work has been the reservoir performance, which we probe here optically. For this reason, we decided to focus on analyzing the reservoir performance by optical read-out with Kerr microscopy. For research purposes, it is advantageous for a demonstrator, as we have full access to the skyrmion position at any time (which is not the case when only one or few MTJs are stacked on top of the device), giving us more data for device performance analysis. To determine the skyrmion position from the Kerr microscopy images, we use the U-Net machine learning: this is a convolutional neural network, which is solely used to binarize the gray-scale Kerr-data (magnetization up/down) to obtain pixel-wise segmentation. The U-Net neural network is described in detail in our publication [Labrie-Boulay, I. *et al.*, *Phys. Rev. Appl.* **21**, 014014 (2024)]. The segmentation is then used to produce an artificial MTJ signal. The U-Net is not part of the reservoir performance analysis but mere a useful tool for us as it is more powerful than conventional Kerr microscopy image analysis (contrast enhancement, Gaussian filters, thresholding etc.) to determine the Skyrmion position.

We hope that this answers the questions of the reviewer. In our revised manuscript, we address this point in the conclusion in detail.

3. (Major question) The current manuscript only takes a specific set of physical parameters (e.g., temperature, amplitude (and frequency) of OOP dc (ac) fields, input voltage amplitude, etc.) to do the gesture recognition. Figure 3, after possible modifications according to above two major questions, although may be a sound demonstration of the ability of this skyrmion reservoir, the overall manuscript to me still lacks sufficient comparisons and surveys of different parameter regimes that can make it a more informative publication in Nature Communications. I separate this into sub-questions below.

We thank the referee for the comments. We answer the questions below in detail. We also added a new paragraph in the results section for future readers.

(3a) Temperature: The temperature-induced Brownian motion of skyrmion is beneficial to

overcome the pinning as mentioned by authors. Then, in the depinning regime, what is the effect of temperature? I would expect higher temperatures cause more randomness or noises of skyrmion motion thus decrease the recognition accuracy, is it correct empirically?

Temperature changes the dynamics of the skyrmion. Since we are limited by the optical resolution of our MOKE readout, we do not change the temperature. For our particular stack, higher temperatures result in smaller skyrmions, making them harder to detect by our optical Kerr microscopy method. Instead of varying the temperature, we adjust the dynamics of the skyrmion by adding out-of-plane (OOP) oscillating magnetic field excitations, which is a more universal method (see answer to question 3c and our reference [Gruber, R. *et al.*, *Advanced Materials* **35**, 2208922 (2023)]).

In the measurements conducted, higher accuracies were achieved when the timescales of the reservoir matched the input. This suggests that a change in temperature is a possible variable that can influence the performance of the reservoir when an optical readout is no longer used. However, by additionally changing the OOP excitation, we can tune the dynamics to any timescale needed. This is explained in our revised manuscript.

(3b) Amplitude of OOP dc field may change the size of skyrmions. Is there any effect of skyrmion size on its recognition ability?

The amplitude of the OOP DC field changes transiently the size of the skyrmion. Here a value is used that optimizes MOKE detection. Since the size of the skyrmion impacts its dynamics, this adjustment also influences the detection accuracy. However, there are obvious paths to obtain more robust skyrmion sizes for instance using synthetic AFM skyrmions [Dohi, T. *et al.*, *Nat. Commun.* **14**, 5424 (2023)]. We have not explored the full size dependence for the performance and this goes beyond the scope of our current work but will be explored in the future.

(3c) Amplitude/frequency of OOP ac field: In the beginning part of the manuscript, pages 2 and 3, authors already emphasize on the advantage of tuning the diffusion timescales by the OOP ac field to avoid energy-consuming rescaling of the dynamical timescales of hand gestures. However, in the main result there is no evidence at least to qualitatively support their claim of this advantage via, e.g., comparing recognition results using different amplitudes/frequencies of OOP ac field while fixing the same time rescaling. I think future readers will expect to see this kind of comparison after they read those statements in pages 2 and 3, and it will give a coherent and convincing proof of authors' claim.

We thank the reviewer for raising this point. We have now investigated the skyrmion diffusion for different AC excitations in detail. The oscillating magnetic field significantly influences the dynamics of the skyrmion and, consequently, the timescales of its movement. A priori if the input timescale is not changed, an increased dynamics or diffusion introduces more noise, leading to reduced detection accuracy. However, if we adjust the input timescales to match the faster response of the skyrmion when exposed to the AC out-of-plane (OOP) magnetic field we obtain a robust detection. So, the AC excitation is a means to tune the timescale of the response of the reservoir and adjust that to the external signal timescale.

We have optimized the timescales of our device to closely align with the dynamics of hand gestures. However, our primary constraint is the Kerr microscopy readout, which operates at a maximum of 16 frames per second, limiting our readout speed. The use of a physical magnetic tunnel junction instead of a simulated one could potentially enable operation at different timescales, making it viable for capturing time series data with a broad range of temporal resolutions.

For future readers, we emphasize this point again in the conclusion of the revised manuscript.

(3d) Input voltage amplitude: Authors have clearly mentioned the skyrmion might be pinned in middle or annihilated at contact regions if the input voltage is too small or too large, respectively.

Then in the voltage range without skyrmion pinning and annihilation, is there correlation between recognition accuracy and input voltage empirically?

We thank the referee for this comment. Certainly, there is a correlation between detection accuracy and input voltage, as our reservoir does not allow for immediate depinning of skyrmions with current. At the current densities used in our work, skyrmions consistently remain in the creep regime. Therefore, higher currents will facilitate faster motion, resulting in enhanced detection accuracy. A future step involves enhancing edge repulsion to allow for larger currents without annihilating the skyrmion. This can be accomplished by modifying the local anisotropy through focused ion beam irradiation at the corners. This work is ongoing and will be explored in a separate study.

Although new experiments are not mandatory since they may require long times for authors to revise this work, for all or some of the parameters listed above, could authors at least provide some qualitative argument or description about their effects on the recognition results?

We hope that with the explanations above and the revised manuscript enough information is provided for the reader to understand the impact. If the referee has further concrete suggestions, we are happy to hear about them and will gladly implement them.

4. (Minor question) Some minor sub-questions are in order.

(4a) While two of the contacts are applied by the input voltage difference, there seems no description about the third contact. Is it grounded?

We realized that we did not clarify, thank you for pointing that out. Contact C is floating and therefore not connected to anything. The contact could allow another input feed for higher complexity of the system to improve detection accuracy. But in the current state there are no currents flowing into this corner.

We added this information in the revised manuscript.

(4b) In page 5, the third paragraph, the sentence “It is also visible that the non-linear transformation to improve the skyrmion response does not change the accuracy significantly for the pure software-based approach.” is not clear. What does it mean? What is the non-linear transformation?

We thank the referee for the comment. The sentence is indeed unclear without proper context. We clarify this in the revised manuscript:

The intensity signal from each gesture must be converted into a voltage suitable for application to the reservoir. To prevent high peaks from potentially causing annihilation of the skyrmion within the corners, we apply a hyperbolic tangent function to flatten these peaks. Additionally, we have trained a support vector machine on the transformed input to confirm that this non-linear transformation does not significantly affect performance compared to the pure software-based reference approach.

We have updated the manuscript by relocating this statement to the appropriate section in the methods, where we provide further elaboration.

5. (Minor question) In the Results section and Methods section, authors describe their material. Following information may be better to add for future readers. (i) In which (or both?) of the ferromagnetic CoFeB or MgO layers that skyrmion lies? (ii) What is the typical skyrmion size in this system (for the specifically applied OOP dc field)? (iii) How does this system induce spin-orbit torque by current? I suggest adding information of these even though they must be trivial to experts in this field.

We are pleased to address the questions and incorporate the following information into the manuscript:

(i) Only the CoFeB layer exhibits ferromagnetic properties, with MgO used to induce perpendicular magnetic anisotropy in the CoFeB layer. We have refined the wording in the "Sample Preparation" section to prevent any further misconceptions.

(ii) The typical size of skyrmions within this stack is optimized for Kerr microscopy readout, typically measuring about 4-5 micrometers. This size can vary depending on temperature and the applied out-of-plane (OOP) DC magnetic field. This detail has been added to the "Sample Preparation" section.

(iii) The bottom Ta layer in direct contact with CoFeB induces spin orbit torques within the magnetic layer. This information has also been included in the "Sample Preparation" section.

6. (Minor question) In the section of Data Evaluation, the procedure of gathering the dataset may be described in a clearer way. I suggest some modifications as follows for better understandings by future readers.

(i) For instance, a simple schematic figure showing the person's hand doing gesture together with the radar system may be good, since to me it is not clear that, in the sentence "... showing the angle or the amplitude against the relative velocity and the distance to the radars", what the angle and amplitude actually mean? In this schematic figure the distance/angle could be clearly indicated.

Yes, the given dataset was not deeply elaborated on, as it was already used in another publication [Kreutz, F., *et al.*, in *2021 7th International Conference on Event-Based Control, Communication, and Signal Processing (EBCCSP)* 1–4 (2021).]. To reduce possible misconceptions, we added the requested schematic figure as "Figure SUP1".

(ii) If possible, perhaps authors could provide more details in this figure and/or in text of what data they measure in these experiments to do the Fourier transform.

We are pleased to provide further details on the Fourier transformation. The used sensor is a frequency-modulated continuous wave radar employing a chirp-sequence of 64 chirps per frame, with 32 sample points per chirp [Kreutz, F. *et al.*, in *2021 7th International Conference on Event-Based*

Control, Communication, and Signal Processing (EBCCSP) 1–4 (2021)]. The data of just one sensor and one chirp is used to achieve the Range-Doppler map via two consecutive Fourier transformations [Gamba, J. *Signals and Communication Technology*. Springer Singapore, Singapore, 2020]. For the Range-Angle maps, the phase difference relative to a second sensor is utilized. Further details on the Range-Angle maps in [Stephan, M. *et al.*, in *25th International Conference on Pattern Recognition*, page 8, Milano, Italy, 2021.].

To improve accessibility of the paper, we also added these details to the methods section of the revised manuscript.

(iii) In Fig. SUP1, the color bars should be shown.

We appreciate the feedback. We have included the missing color bar and adjusted the axes to enhance clarity and intuitiveness for better understanding.

7. (Minor question) Some typos are as follows:

(i) In page 3, the 7th row of the last paragraph, the sentence is a bit confusing, "... is realized by..." may need to be modified as "... as realized by..."

(ii) In caption of Fig. 2, 1st row, "... a with..." should be modified to "... with a..."

(iii) In caption of Fig. 3, "... b) Comparison detection..." might need to be modified as "... b) Comparison of detection...". Also, it is better to add an abbreviation such as "... reservoir (RSVR)..." in this caption.

We thank the referee to point-out the typo mistakes. We have corrected the mentioned passages in the revised version of the manuscript.

Reviewer #2 (Remarks to the Author):

In this manuscript, the authors present gesture recognition using reservoir computing powered by the current-induced dynamics of skyrmions. In particular, the authors claim that when the time scale of input signals is close to that of skyrmion dynamics, the performance of the gesture recognition task is comparable to that of state-of-the-art software-based neural networks, without the need for preprocessing of data.

The manuscript is clearly written and refers to previous literature appropriately. In addition, the quality of the data and presentation are satisfactory. However, I believe this paper does not meet the high standards of Nature Communications for the following reasons.

We thank the reviewer for taking the time to conscientiously review our article. While referee #1 recommends publication of a suitably revised manuscript, this referee has raised a number of points that we address in the revised manuscript to make our work appropriate for publication in Nature Communications. Below, we provide further details on why we believe our revised manuscript meets the requirements for publication.

1. The first point concerns the significance of this paper. So far, experiments on reservoir computing using skyrmion dynamics have been reported by several groups [for example, K. Raab, et al., *Nat. Commun.* 13, 6982 (2022), T. Yokouchi et al., *Science Advances* 8, abq5652 (2022), and O. Lee et al., *Nat. Mat.* 23, 79 (2023)] (one is the author's group). While the detailed structure differs in each paper, the key concept of applying skyrmion dynamics to physical reservoir computing has already been established by these works. Hence, the main significance of this paper lies in the demonstration of physical reservoir computing without the need for preprocessing of data.

However, it seems obvious that preprocessing is not required when the time scale of input data matches that of the reservoir system. Of course, proving the idea experimentally is quite important. However, it is questionable whether this paper will attract much attention from general audiences in a broad field, and thus a more specific journal might be more appropriate.

We acknowledge the relevance of the papers mentioned and recognize the significant research contributions they have made. However, we believe that our concept and the data we have utilized distinguish our work and represent a significant advancement beyond the scope of existing publications.

While the device utilized in Ref. [K. Raab, *et al.*, *Nat. Commun.* **13**, 6982 (2022)] closely resembles our setup, which originates from our own group, the operational methods differ significantly. In the referenced paper, a space-multiplexed readout is employed with constant voltages applied, alongside multiple simulated MTJs that are time-averaged for readout purposes. This setup results in quasi-static skyrmions compared to our approach, which exploits real-time dynamics. Our time-multiplexed reservoir computing using skyrmions has so far only been suggested theoretically [e.g. R. Msiska, *et al.*, *Advanced Intelligent Systems* **5**, 2200388 (2023)]. The mentioned publication has demonstrated proof-of-concept for Boolean logic using binary inputs, but real-life data detection has not been achieved.

The second publication mentioned, [T. Yokouchi *et al.*, *Science Advances* **8**, abq5652 (2022)], represents the only other approach tested with real-life data. However, it requires 41 devices in parallel, and input is achieved through a changing magnetic field, which poses challenges for scalability (field generation of the required field strengths exhibits poor scaling). Furthermore, it does not have the flexibility to deal with data at different timescales. For realization in a device, our approach can work with MTJs, while the proposed approach in this paper uses imaging or Hall effect measurements that are both not easily implemented for fast and easy read-out. This renders our device highly CMOS-compatible, while still demonstrating remarkable performance with a simple layout comprising of just one device.

In the last publication, [O. Lee *et al.*, *Nat. Mat.* **23**, 79 (2023)], relies solely on synthetic data and employs changing magnetic fields for input. In combination with its output necessitating a vector network analyzer, it renders the device not scalable and requiring advanced postprocessing. Moreover, the material used is not CMOS-compatible, necessitating cooling below 50 K and, in some cases, down to 4 K for evaluation.

In summary, while previous approaches offer valuable insights into reservoir computing with skyrmions, our current approach stands out as it presents a CMOS-compatible device capable of performing with complex real-life data at flexible timescales. Our approach is straightforward to implement in future devices, offering easy steps to enhance performance and adjustability to specific timescales for various applications. Additionally, scalability to the nanoscale is easily achievable with our device design.

2.The second point is about the conclusion of this work. The authors conclude that skyrmion reservoir computing is competitive and can even exhibit superior performance compared to state-of-the-art software-based neural networks, a result shown in Fig. 3. However, the error bars for the skyrmion-based reservoir are quite larger than those for the state-of-the-art software-based neural network (Fig. 3). This indicates that the performance of the skyrmion reservoir depends on the choice of data set for K-fold training. Does this imply that the performance of the skyrmion-based reservoir is less than that of the state-of-the-art software-based neural networks?

In addition, several error bars exceed the accuracy of 100 %. What does an accuracy above 100%

mean? It would be better to show all data points instead of presenting the average value and errors or to reconsider the calculation method of error bars.

We apologize that we did not make this sufficiently clear and have now revised our manuscript to show where our approach excels compared to conventional approaches. We next split our answer into the individual questions to provide details for the individual points:

Does this imply that the performance of the skyrmion-based reservoir is less than that of the state-of-the-art software-based neural networks?

As a major change compared to the previous manuscript, we have now carried out an additional study that correctly compares the performance of the reservoir with the linear SVM (RSRV + LIN SVM) and the linear SVM on the full data (LIN SVM). While previously we did use a larger data set for (LIN SVM) than for (RSRV + LIN SVM) because of the longer time it takes to carry out the experiments for the reservoir case, we now use identical data sets for both for a robust comparison. Using the same data sets and calculating the errors correctly we provide the following key finding: we find that (i) the reservoir with a single MTJ and with a linear SVM (RSRV (1 MTJ) + LIN SVM) outperforms the linear SVM (LIN SVM) and (ii) that by adding more MTJs we can significantly improve the performance of the reservoir so that a reservoir with e.g. 5 MTJs and a linear SVM (RSRV (5 MTJ) + LIN SVM)) outperforms even the advanced and energetically much more costly radial basis function SVM (RBF SVM) with the difference being clearly outside the error bars. Adding more MTJs provides the linear readout layer with more information about the skyrmion position explaining the increased performance. This results in increased performance across all gesture pairs.

To validate the increased energy consumption of the software-based approach, we performed benchmarks of the output layer, detailed in the revised Supplementary Material.

In addition, several error bars exceed the accuracy of 100 %. What does an accuracy above 100% mean? It would be better to show all data points instead of presenting the average value and errors or to reconsider the calculation method of error bars.

Errors were determined using the standard deviation. Due to the increased fluctuation, as discussed in the previous answer, this can lead to errors exceeding 100%, although performance cannot surpass 100% itself. To enhance clarity and facilitate comparison between the different approaches, we now

present the mean and standard error as a more robust visualization method instead of representing each value with ten individual points. Consequently, no error now exceeds 100%.

REVIEWERS' COMMENTS

Reviewer #1 (Remarks to the Author):

From the new version of manuscript and rebuttal, it can be seen that the authors have seriously considered my comments and done a thorough work to answer all of my major and minor questions. In the previous review, my most crucial questions were those in comment 1, for which now authors have provided a correct and fair renewed comparison between the skyrmion reservoir and SVM based on the same number of datasets, and their new Figure 3 now unambiguously shows the better performance by skyrmion reservoir than that by SVM. Although the averaged accuracy in page 2 of their rebuttal for skyrmion reservoir seems to be a merely mild improvement by less than 10% compared to RBF SVM, it should be stressed that in the supplement authors also provide the comparison of energy cost or operation speed between these systems and show that skyrmion reservoir can gain at least one order of magnitude of the time cost (or energy cost) compared to RBF SVM. I believe these results provide a promising and solid demonstration of the potential of skyrmions as candidates for next-generation spintronics reservoirs in machine learning. I still have following minor comments that need authors to further answer, but overall I recommend the publication of this manuscript in Nature Communications.

Regarding my previous comment 1:

(1a) After checking the authors' new data and reply, now it is clear that the skyrmion reservoir outperforms SVM. For this comment, I would like to ask if authors could incorporate the right figure (average of mean accuracy) in page 2 of their rebuttal into Fig.3 in main text [maybe as an inset in Fig.3(b)], since it provides an additional clear comparison. For future readers, this result may seem to be a merely mild improvement by using skyrmion reservoir than RBF SVM, since the accuracy only improves by around 10% on average; therefore, perhaps authors can try to stress on the result of inference time comparison in the main text.

(1b) Could authors explain what exactly "inference time" is? Is it the "training time" for the readout layer or SVM? I could not find the definition of it in the new version of manuscript. This might be a trivial question but it may confuse future readers. Once this is clarified, this comment is convincingly answered by authors who have shown that the skyrmion reservoir can at least gain one order of magnitude of the inference time than RBF SVM which directly demonstrates its advantage of lower power consumption.

(1c) There is no problem in my opinion for authors who unintentionally put incorrect analysis in a manuscript. It is reviewers' job to detect the possible mistakes. The new Fig.3 has been corrected by a fair comparison taking the same size of dataset respectively for skyrmion reservoir and SVM,

and now the result becomes valuable and trustworthy, showing unambiguously the advantage of skyrmion reservoir. No further questions.

For other comments, authors have answered them well and I have no further questions. Some typos: In Data Evaluation section in page 8, two sentences including “Figure SUP1” should be modified as “Figure 5”. It seems authors changed some figure numerations in main text from “Figure SUP xxx” to “Figure xxx”, but some text sentences still quote previous “Figure SUP xxx”. Please check if there are any other inconsistencies to prevent confusion.

Reviewer: Mu-Kun Lee (Waseda University, Japan)

Reviewer #2 (Remarks to the Author):

I appreciate the authors' sincere response to my comments. The authors have addressed all my concerns and suitably revised the manuscript. In particular, in the previous manuscript, the benefit of the reservoir was not clear. However, in the revised manuscript, the authors added new experiments, which clearly indicate the reservoir improves the performance of gesture recognition tasks. Thus, I recommend the publication of the manuscript in Nature Communications.